# A Hybrid Modeling of the Physics-Driven Evolution of Material Addition and Track Generation in Laser Powder Directed Energy Deposition

**DOI:** 10.3390/ma12172819

**Published:** 2019-09-02

**Authors:** Gabriele Piscopo, Eleonora Atzeni, Alessandro Salmi

**Affiliations:** Department of Management and Production Engineering (DIGEP), Politecnico di Torino, Corso Duca degli Abruzzi 24, 10129 Torino, Italy (G.P.) (E.A.)

**Keywords:** directed energy deposition, thermal analysis, additive manufacturing, finite element model, laser powder deposition

## Abstract

Directed Energy Deposition (DED) is one of the most promising additive manufacturing technologies for the production of large metal components and because of the possibility it offers of adding material to an existing part. Nevertheless, DED is considered premature for industrial production, because the identification of the process parameters may be a very complex task. An original hybrid analytic-numerical model, related to the physics of laser powder DED, is presented in this work in order to evaluate easily and quickly the effects of different sets of process parameters on track deposition outcomes. In the proposed model, the volume of the deposited material is modeled as a function of process parameters using a synergistic interaction between regression-based analytic models and a novel element activation strategy. The model is implemented in a Finite Element (FE) software, and the forecasting capability is assessed by comparing the numerical results with experimental data from the literature. The predicted results show a reasonable correlation with the experimental dimensions of the melt pool and demonstrate that the proposed model may be used for prediction purposes, if a specific set of process parameters that guarantees adequate adhesion of the deposited track to the substrate is introduced.

## 1. Introduction

Additive Manufacturing (AM) technologies are recognized as the future of the manufacturing industry, thanks to their possibilities, in terms of shape design, part functionality, and material efficiency [1,2,3,4,5]. Directed Energy Deposition (DED) is an emerging AM process that shows great potentialities. DED has the ability to process large build volumes greater than one cubic meter, to deliver the material directly into the melt pool, to repair existing parts, and to change the material composition during the building process, thereby creating functionally-graded materials [3,6]. Considering a laser as a thermal source and powder as a feedstock material, several DED technologies, such as Laser Metal Deposition (LMD), laser cladding, Laser Engineering Net Shaping (LENS^®^), Directed Light Fabrication (DLF), and Direct Metal Deposition (DMD™), have been developed [7,8,9,10]; the term Laser Powder Directed Energy Deposition (LP-DED) is used hereafter in the paper to identify this category of processes.

At present, the LP-DED technology is considered premature for industrial production [7]. One reason for this is that, to ensure the quality of the fabricated part, it is necessary to identify the optimal combination of process parameters for each material and geometry to be fabricated. However, process parameters in LP-DED are generally selected using a trial-and-error approach, based on the users’ experience, a process that is very expensive in terms of cost and time. Thus, research efforts have been focused on developing numerical modelings of the LP-DED process, in order to allow a reduction in the number of experiments required to determine the optimal process parameter configuration and, consequently, the time and the cost needed to identify a process window in which the process is stable. In the LP-DED process, the key point of the process is the generation of the melt pool, since the melt pool constitutes the initiation of the deposited track [7,8]. Therefore, modeling efforts reported in the literature have specifically been devoted to simulating the melt pool and track generation sometimes without considering the material addition. In most cases, the scientific understanding of welding is translated into an LP-DED process, and on occasion, some major phenomena and characteristics that are specific to the AM process are neglected.

In this work, at first, an extensive literature review of the state-of-the-art of LP-DED is conducted in order to shed light on the physics of the LP-DED process. Then, the results of this analysis are exploited in order to develop a hybrid physics-driven model to simulate the deposition of a single track. Thermal phenomena are solved using the finite element method, introducing the material addition by implementing an analytic model. In detail, analytic relationships linking the dimensions of the track and the process parameters are identified by regression analyses, based on experimental data. Moreover, in order to develop a smart simulation tool, which can improve the industrial readiness for LP-DED, a specific meshing strategy is adopted to allow reliable results to be obtained in a reasonable computational time. The model is validated against experimental data taken from the literature for 316L stainless steel, and it is proven to be a viable tool to predict track adhesion onto the substrate and to identify suitable parameters of the deposition process.

## 2. Analysis of the LP-DED Process

The physical effects involved in the LP-DED process occur at different levels, from microscopic to macroscopic. For this reason, different modeling approaches are used in the simulation of AM processes [11,12]. They differ in the characteristic dimension of the simulated phenomena and in the outputs of the model, from the melt pool (micro-) to the single layer (meso-) to the part (macro-scale).

### 2.1. Physics of the LP-DED Process

Numerous process parameters directly influence the material properties, part geometry, and tolerances of components fabricated by means of LP-DED, but the relationships between the process conditions and the final properties are not easy to understand, as the phenomena involved in the LP-DED process are very complex. Figure 1 shows the main parameters that affect the LP-DED process, grouped into five categories. The parameters that are considered the most relevant in the literature are highlighted in bold. It is possible to identify three main mechanisms in the LP-DED process: the powder stream process, melt-pool generation, and solidification of the track [7,8].

#### 2.1.1. The Powder Stream Process

In LP-DED, a nozzle is used to generate the powder stream by means of the carrier gas. The phenomena involved in the powder stream process refer to the powder-flow distribution, the laser–powder interaction, and to the carrier gas flow. Two main configurations of the powder-fed nozzle are available in existing systems, that is lateral and coaxial nozzle feeding. LP-DED systems were initially based on a lateral nozzle, but the characteristics obtained using this nozzle configuration were highly direction dependent [13,14]. To overcome this direction dependency, coaxial nozzle feeding was developed, and it is currently the most commonly-used solution [10,13,15]. The powder particles are transported to the substrate by means of the carrier gas, which is generally argon [16]. The carrier gas has two main objectives: it allows a stable powder flow to be obtained, and it protects the fusion zone from oxidation. Gibson et al. [16] showed that the quality of the produced part decreases as the gas flow rate increases, due to the greater disturbance in the powder flow induced by the higher gas flow rate. Pinkerton and Li [17] and Ibarra-Medina and Pinkerton [18], using coaxial nozzle feeding, demonstrated that three different patterns of the powder flow can be distinguished during the flight time. First, an annular distribution of the powder stream is observed near the nozzle tip. Second, the powder stream converges, and the maximum concentration is reached at the merging point. In this second phase, the distribution of the powder flow is of a Gaussian type. Third, the powder flow diverges after the merging point, while maintaining its Gaussian distribution. During the powder stream process, the in-flight powder particles interact with the laser beam, close to the nozzle tip, and a temperature increase of the powder is observed [13]. This increment in temperature is related to the absorption, reflection, and scattering effect of the powder (Figure 2), and as a consequence, the laser power is reduced. The laser attenuation in coaxial LP-DED can be as much as 50%, as observed by Lin [19]. Finally, the reduction in laser power depends on the powder material, powder concentration, powder feed rate, scanning speed, and laser power [19,20,21]. Under specific conditions, the powder can melt in-flight, but it is often preferred to avoid this [16].

#### 2.1.2. The Melt Pool Generation

The energy provided by the laser generates a melt pool on the substrate. Simultaneously, the powder particles are fed into the melt pool. When the powder enters the melt pool, it melts very rapidly and generates the deposited track. Three main process parameters influence the melt pool, namely the laser power, the powder mass flow rate, and the traverse speed [8]. However, the physics of the process cannot be described accurately using only these three parameters. Qi et al. [21] highlighted more than ten parameters that may influence the characteristics of the deposited parts. These parameters are: laser power, laser characteristics (wavelength, mode), laser spot size, carrier gas flow (temperature, pressure, velocity), powder material properties (size and distribution, shape, thermal properties), nozzle configuration, and process characteristics (hatching distance, scanning speed, scanning strategy). Huge temperature gradients are produced during melt pool generation. These gradients cause high surface tension and free convective and Marangoni flows [7]. Moreover, since high temperatures are reached, the thermal radiation of the molten material plays a fundamental role in the process. Heat conduction through the substrate and the convection generated by the shielding gas are another two phenomena that are involved in melt pool generation (Figure 2).

#### 2.1.3. The Solidification of the Track

The solidification mechanism of the liquid phase is mainly governed by conduction through the substrate and by convection. This solidification mechanism is responsible for the main important characteristics of the solid track, including the resulting microstructure and the residual stresses. The microstructure of the deposited track is essentially influenced by two parameters: temperature gradient *G* and solidification growth rate *R* [22,23,24]. The G/R ratio determines the solidification structure, and if the ratio decreases, the microstructure evolves from planar to cellular to columnar to equiaxed dendrites. The product of *G* and *R*, that is the cooling rate, defines the dimension of the resulting microstructure. If a higher cooling rate is obtained, a finer microstructure is produced [13]. The high-temperature gradients and the rapid thermal history in the LP-DED process inevitably determine the presence of significant residual stresses in the produced component.

Hofmeister et al. [25] developed one of the first finite element models to analyze thermal behavior during the fabrication of a thin 316 stainless steel wall produced by the LENS^®^ technology. Only heat conduction was considered in this model; in addition, the temperature of the substrate was kept constant at 300 ${}^{\circ }C$, and the elements were activated at the melting temperature. In the three-dimensional model proposed by Hu and Kovacevic [26], the thermal power was controlled and varied throughout the simulation in order to maintain a predefined melt pool width and a constant wall thickness. In this model, all the elements of the analyzed track were activated at the beginning of the analysis, and the addition of material was not considered. Moreover, the convective and radiation effects were evaluated using a lumped heat transfer coefficient, an assumption that was derived from the welding process [27]. The effect of latent heat was simulated by modifying the specific heat capacity at the melting temperature. This modification of the specific heat capacity was derived from the work of Brown and Song [28], who focused on modeling the welding of large structures.

### 2.2. Modeling of the LP-DED Process

Labudovic et al. [29] developed a three-dimensional thermomechanical model of Direct Metal Powder Deposition (DMPD) in order to evaluate the temperature profiles, the dimension of the melt pool, and the distribution of the residual stresses. In this model, a lumped heat transfer coefficient, a Gaussian distribution of the thermal load, and the modified specific heat were considered, as in the work of Hu and Kovacevic [26]. The effect of convective motion within the melt pool was neglected. Toyserkani et al. [30,31] evaluated the influence of the process parameters and laser pulse shaping parameters on the deposited geometry. In their three-dimensional model, a Gaussian distribution of the heat source and of the modified specific heat was assumed. Moreover, the radiative and convective conditions were combined in a lumped heat transfer coefficient. Costa et al. [32] coupled the thermal model with phase transformation kinetic data in order to obtain the hardness distribution of AISI 420 steel. In their model, a Gaussian distribution was assumed for the heat source, and the latent heat was taken into account using an internal heat source term. Wang et al. [33] simulated the temperature distribution and the phase evolution in a thin wall produced in SS410. The same authors demonstrated that, in order to obtain a constant melt pool dimension in the process, the laser power must be decreased for each consecutive deposited layer. Peyre et al. [34] developed a three-step analytical and numerical model to predict the wall morphology and the thermal distribution during the deposition of a thin Ti6Al4V alloy wall. In this thermal model, the track was activated at the beginning of each step, and the addition of material was simulated by means of an analytical modification of the thermal conductivity, which was assumed to be very small after the melting front. Fallah et al. [35], using a transient finite element model, simulated the thermal distribution and real-time formation of the melt pool. In their model, the distribution of the heat source was considered as Gaussian, and the latent heat was taken into account through a modified specific heat capacity at the melting temperature. The simulated temperature distribution showed an error of 5% in comparison with the experimental data. The simulated height was predicted with an average error of almost 8%.

Neela and De [36] investigated the effect of process parameters on thermal cycles, resulting from the deposition of a single track of stainless steel. In their three-dimensional finite element model, the addition of material was simulated using the deactivation and activation features of Abaqus 6.7. The activation was performed using a time-step algorithm, and the dimensions of the activation volume were independent of the process parameters. Ahsan and Pinkerton [14], using a coupled analytical-numerical model, studied the thermal history, the track profile, and the microstructure scale during the deposition of a single track of Ti6Al4V alloy on a substrate of the same material. In their work, in order to simplify the analytical model, the substrate was assumed to be a semi-infinite plane, the material properties were temperature-independent, and the heat losses through convection and radiation were ignored. Manvatkar et al. [37], who developed a three-dimensional finite element model by means of Abaqus 6.8 EFI, analyzed the thermal cycle and melt pool width during the deposition of a thin 316L stainless steel wall. They showed that, for constant laser power and laser traverse speed, an increase in the number of deposited layers resulted in an increase in the layer width and in the maximum temperature value and that a lower cooling rate was obtained. Amine et al. [38] used Abaqus to evaluate heating and reheating cycles resulting from the deposition of a thin multilayer 316 L stainless steel wall produced by the LENS process. They showed that, by increasing the laser power and/or reducing the scanning speed, higher temperature and cooling rate values, as well as higher dimensions of the melt pool of the re-melted layers were obtained. Chiumenti et al. [39] developed a framework for the thermal numerical simulation of the LP-DED process. In their work, the temperature distribution was analyzed and the sensitivity of the model, with respect to the energy absorption and dissipation mechanism, was presented. They showed that the temperature was influenced to a great extent by the absorption coefficient and that the radiation was the main dissipation mechanism in the surrounding environment. Shah et al. [40] used a commercial FE package to study the effect of the powder mass feed rate and of the duty cycle of the laser on the residual stresses of Inconel 718 deposited on a Ti6Al4V substrate. They showed that residual stresses may be reduced using a short laser pulse length. Moreover, they showed that a high powder mass feed rate produced high stress values, especially near the starting edge of the deposited track. Gan et al. [22] developed a numerical model in order to study the thermal behavior, fluid flow, melting-solidification, and multicomponent mass transfer during the deposition of a Co-base alloy on a 38MnVS carbon steel substrate medium.

An analysis of the existing models, which is summarized in Table 1, shows that most of the works were focused on the investigation of the temperature distribution in the deposited track and in the substrate, as well as on the analysis and prediction of the deposited geometry. Some basic assumptions, such as the combined heat transfer coefficient and the modification of specific heat in order to model the latent heat, were based on the physics of the welding processes. However, even though there are similarities between LP-DED and welding, the additive process introduces more complexity and challenges. In fact, the scale of the process is much smaller; higher temperatures are reached; and solidification and cooling are very rapid. Owing to the high surface temperatures at the melt pool, radiation and convection become non-negligible. Moreover, from the analysis of the literature, it emerges that one of the most challenging issues in the modeling of the LP-DED process is the way in which deposited material is added to the model and the definition of its dimensions. In general, two different approaches can be used: the quiet element method and the inactive element method. In the quiet element method, all the elements are included in the solution domain at the beginning of the simulation. The material addition is simulated using a subroutine that modifies the thermal and elastic properties of the elements, such as thermal conductivity or Young modulus, by means of scaling factors. In such a way, the element switches from quiet to active. This method is very easy to implement. However, the results depend on the scaling factor values, and furthermore, convergence problems may occur [41,42]. In the inactive element method, the elements are only included in the solution domain when they are added to the model according to the physics of the process. In this method, the equations are only solved for the nodes of the active element, and this may lead to a reduction in the calculation time [42]. Nevertheless, this method has not yet been fully implemented in commercial software, and some discontinuity problems may occur at the nodes shared by active and inactive elements. Thus, up to now, the addition of material has usually been simulated by means of a fictitious material, or it is even ignored, and the dimensions of the track are assumed as being independent of the process parameters.

## 3. The Hybrid Analytic-Numerical Model of the LP-DED Process

The modelization of the process is presented in detail in the following sections, in terms of heat transfer, energy source, and addition of deposited material.

### 3.1. Heat Transfer Modeling

In the LP-DED process, the heat conduction through the substrate mainly governs the melt pool dynamics, together with convection due to Marangoni flows and the convection generated by the carrier gas. As a result of the high surface temperature, thermal radiation is also very important. In the following sub-sections, it is described how these thermal phenomena were implemented in the model.

#### 3.1.1. Heat Conduction

In the model, the transient temperature distribution is calculated by solving the general heat conduction equation:(1)$$\rho c_{p}\frac{\partial T}{\partial t}=-\nabla q+Q$$
where *Q* is the internal heat source in $W/m^{2}$, q is the heat flux vector in $W/m^{2}$, ρ is the material density in $kg/m^{3}$, and $c_{p}$ is the specific heat in J/(kg·K). The heat flux is related to the temperature gradient by Fourier’s law:(2)q=−k∇T
where *k* is the thermal conductivity of the material in $W/(m^{2}\cdot K)$ and ∇T is the thermal gradient. Combining Equations (Equation 1) and (Equation 2) leads to:(3)$$\rho c_{p}\frac{\partial T}{\partial t}=\nabla \left(k\nabla T\right)+Q$$
which can also be expressed as:(4)$$\rho c_{P}\frac{\partial T}{\partial t}=\frac{\partial }{\partial x}\left(k\frac{\partial T}{\partial x}\right)+\frac{\partial }{\partial y}\left(k\frac{\partial T}{\partial y}\right)+\frac{\partial }{\partial z}\left(k\frac{\partial T}{\partial z}\right)+Q$$

In order to solve Equation (Equation 4), the initial and the boundary conditions have to be introduced into the model. The initial condition at the time t=0 for the elements included in the substrate is:(5)$$T\left(x,y,z,t=0\right)=T_{0}$$
where $T_{0}$ is the ambient temperature in K. A specific initial condition is required for the new elements of the deposited track that will be included (activated) at the time $t=t_{act}$ in the computational domain. This condition takes into account the increment of the temperature of the powder during the in-flight time and it is assumed that:(6)$$T\left(x,y,z,t_{act}\right)=T_{act}$$
where $T_{act}$ is the activation temperature in K. The boundary conditions are: (7)$$-k\left(\nabla T\cdot n\right){|}_{\Omega }=\left\{\begin{array}{cc} \beta I_{L}\left(x,y,z,t\right)-h_{c}\left(T-T_{0}\right)-\varepsilon_{r}\sigma \left(T^{4}-T_{0}^{4}\right) & \mathrm{if}\;\Omega \in \Omega_{L}\\ -h_{c}\left(T-T_{0}\right)-\varepsilon_{r}\sigma \left(T^{4}-T_{0}^{4}\right) & \mathrm{if}\;\Omega \notin \Omega_{L} \end{array}\right.$$
where n is the vector normal to the heat exchange surface Ω, $\Omega_{L}$ is the surface irradiated by the laser beam, β is the correction coefficient of the heat source, $I_{L}\left(x,y,z,t\right)$ is the laser intensity distribution in $W/m^{2}$, $h_{c}$ is the heat convection coefficient in $W/(m^{2}\cdot K)$, $\varepsilon_{r}$ is the surface emissivity, and σ is the Stefan–Boltzmann constant $5.67\times 10^{-8}$
W/($m^{2}\cdot$K^4^).

The material properties such as the density ρ(T), the specific heat $c_{p}\left(T\right)$, and the thermal conductivity k(T) were assumed to be temperature dependent. In order to take into account the convective redistribution of heat in the melt pool due to the thermocapillary flows (i.e., Marangoni flows), a higher value of thermal conductivity was considered, if the melt pool temperature was above the liquidus temperature. According to the experimental work of Lampa et al. [43], a suitable value of the effective thermal conductivity, $k_{a}\left(T\right)$, should be 2.5-times that of the stationary thermal conductivity, k(T), of the material. Thus, in this model, it is assumed that: (8)$$k_{a}\left(T\right)=\left\{\begin{array}{cc} k(T) & \mathrm{if}\;T\leq T_{liq}\\ 2.5k(T) & \mathrm{if}\;T>T_{liq} \end{array}\right.$$
where $T_{liq}$ is the liquidus temperature of the material in K.

Moreover, the introduction of latent heat into the model allows the liquid-solid transformations that occur during the deposition process to be considered. The latent heat of fusion $L_{f}$ is applied between the temperature of solidus $T_{sol}$ and of liquidus $T_{liq}$.

#### 3.1.2. Convection

As detailed in Section 2.1.1, the powder particles are fed into the melt pool using an inert gas. As a result of its turbulent nature, this gas flux causes forced convection at the top of the melt pool and track. Thus, the convection mechanism is considered as a forced convection mechanism. Gouge et al. [44] demonstrated that the value of convection coefficient $h_{c}$ in laser cladding varies as the axisymmetric exponential decay function varies from the center of the laser beam. However, the use of a variable coefficient could increase the calculation time, without significantly affecting the numerical results [44,45]. For this reason, in our model, the value of the convection coefficient was assumed to be constant and was applied to the melt-pool and deposited track, within the influence radius of the carrier gas. Natural convection was applied to the other regions.

#### 3.1.3. Radiation

Thermal radiation becomes very important when the temperature is high, as occurs in LP-DED processes. This heat transfer mechanism was included in the model by means of the surface emissivity coefficient, $\varepsilon_{r}$. A constant value of surface emissivity was assumed in this model.

### 3.2. Energy Source

The heat source, which is a laser beam in the case of LP-DED, provides the energy necessary to melt the substrate and the fed powder, and it defines the thermal field inside the component. The numerical description of the laser power distribution is of crucial importance to obtain reliable results. Michaleris [41], Yang et al. [45] and Heigel et al. [46] used a Goldak distribution to describe the heat distribution that was developed for the welding process [47]. However, the Goldak distribution is not a real representation of the laser intensity, but is a fictitious model of the laser power density that is effective at describing transient temperature fields from a moving heat source, after a calibration procedure based on empirical data [47]. It is important to point out that any change in material or process parameters implies the need for a new calibration of the Goldak distribution.

A measurement of the distribution of the laser intensity at the focus plane in the LP-DED process is shown in Figure 3 [48]. It is possible to observe that the intensity was quite uniform and dropped sharply after a specific dimension that corresponded to the laser beam diameter at the focus plane (laser spot). Accordingly, in this paper, the distribution of the laser beam intensity, $I_{L}$, was modeled as a uniform-distributed volumetric heat source: (9)$$I_{L}\left(x,y,z,t\right)=\frac{P}{\pi r_{L}^{2}H}$$
where *P* is the laser power in W, $r_{L}$ is the radius of the laser beam at the focus plane in mm, and *H* is the height of the deposited track in mm, which in turn depends on the process parameters.

During the process, the laser energy was absorbed, reflected, and scattered by the in-flight metal particles and by the substrate. The value of the power has to be reduced, using the correction coefficient β, to consider these phenomena [19,20,21]. Picasso et al. [49], Huang et al. [50] and Diniz Neto et al. [51] modeled this coefficient using an analytical formulation and showed that the β-value depends on the process parameters, on the material properties, and on the powder size. Unocic and DuPont [20] performed an experimental analysis and demonstrated that, under the analyzed process parameters, the correction coefficient varies between 0.3 and 0.5. In their experiment, the laser power was varied between 125 W to 500 W, the powder feed rate was varied between 80 mg/s to 330 mg/s, and the scanning speed was varied between 5 mm/s to 35 mm/s. They analyzed the deposition of steel powder on a substrate of the same material and the deposition of copper powder on a steel substrate. From their experimental results, it is possible to point out that the correction coefficient is mainly influenced by the deposited and substrate materials. The analyzed process parameters had no or little influence on the value of the correction coefficient. For this reason, in this work, the value of the correction coefficient was assumed to be constant.

### 3.3. Addition of Deposited Material

The addition of deposited material in the model was obtained by adding elements to the computational domain. This mechanism is named as element activation. Elements are activated according to an activation strategy, which in our approach allows taking into consideration the physics of the process, and this represents the core of the present model. Specifically, the activation strategy takes into account the position of the deposition head in order to identify the position of the activation volume, whose dimensions result from the process parameters (i.e., the laser power, the powder feed rate, and the scanning speed). Toyserkani et al. [10] found that the dimensions of the deposited track (width and height) are dependent on the specific energy *E* (J/mm^2^) and the powder density *G* (mg/mm^2^), which are defined as:(10)$$E=\frac{P}{2r_{L}v}$$
(11)$$G=\frac{Q}{2r_{L}v}$$
where *P* is the laser power in W, *Q* is the powder feed rate in mg/s, *v* is the scanning speed in mm/s, and $r_{L}$ is the laser beam radius in mm. The effect of the process parameters on the track dimensions can be obtained by performing a regression analysis on a small set of experimental data. The empirical relationships will be then used to determine the influence of process parameters on track outputs.

The activation strategy also takes into account the powder–laser interaction that occurs during the powder stream process by associating an initial temperature, the activation temperature $T_{act}$, with the activated elements. During the in-flight time, the powder particles are subjected to heating by the laser beam, and as a consequence, their temperature rises. Numerous works have been developed to study the interaction between a laser and a powder flow and to evaluate powder heating [52,53,54]. These works used a lumped capacitance method and demonstrated that the final temperature depends on the process parameters, the material properties, and the in-flight time. Although the powder temperature depends on the process parameters, no analytical formulations are available that allow the increase in the powder temperature during the in-flight time to be calculated easily. In fact, the calculation of the powder temperature is usually performed using CFD simulations, and in this case, an accurate description of the geometry of the deposition head is required [55,56]. From the work of Ibarra-Medina and Pinkerton [18], it is possible to observe that the temperature of the powder particles in the laser focus plane increases by about 10 K for each 100 W increment of laser power. During the LP-DED process, the change in the powder temperature associated with the laser power oscillation is negligible, and thus, a constant value of activation temperature may be assumed.

## 4. FE Implementation of the Hybrid Model

The Finite Element (FE) method was used to solve the heat transfer problem, as described in Section 3, and the model was implemented in Abaqus/Standard 2018 by Dassault Systèmes^®^. Figure 4 illustrates the workflow of the FE model. The FE model allows the motion of the deposition nozzle and the motion of the laser beam to be distinguished. These two positions are coincident for a standard coaxial nozzle, but using this approach, it is also possible to exploit this model when specific scanning movements of the laser beam (wobbling) are superimposed onto the deposition nozzle motion. The position of the deposition nozzle was used to activate the elements (i.e., to simulate the addition of material), and the laser beam position was used to selectively apply the heat source; both of which were achieved by defining a FORTRAN-coded (analytic) model in specific user-defined subroutines.

The activation strategy adopted in the proposed FE model was controlled by the UEPACTIVATIONVOL user-defined subroutine (Section A.1), which is based on the hybrid quiet inactive element method. The customization of this user-defined subroutine allows the progressive deposition of the material to be simulated. UEPACTIVATIONVOL allows users to implement analytic models for elements activation and evolving free surfaces. The spatial coordinates of the deposition nozzle are computed at each time increment, dt, considering the deposition path and the scanning speed. The updated position of the deposition nozzle was then used to calculate what elements will be activated according to the analytic models that describe the deposition dimensions. In other words, the activation algorithm generates a parallelepiped-shaped activation volume. The height and the width of the activation volume are computed according to the analytic models that correlate the tracks dimensions with process parameters. An initial activation temperature is imposed on each new activated element in order to take into consideration powder stream heating during the in-flight time.

The DFLUX user-defined subroutine (Section A.2) is then used to apply the heat source to the model, according to the laser motion. In this user-defined subroutine, the position of the center of the laser spot is calculated at each time increment, dt, according to the laser scanning strategy. The heat flux of the body is calculated using Equation (Equation 9) and is applied to the computational domain included in the activation volume. The thermal distribution is then calculated using the equations described in Section 3.1. The thermal calculation is repeated at each time increment in the step up to time $t_{end}$, which is the time required to complete the deposition.

The proposed thermal hybrid analytic-numerical model was validated against the experimental conditions described in the work of El Cheikh et al. [48]. They fabricated 316L stainless steel single tracks on steel substrates by means of a machine using a deposition head with a coaxial nozzle and a laser with a maximum power of 700 W and a radius at the focus plane of 0.265
mm. Experimental results from the 27 deposition tests were available, and the dataset was partitioned as follows:Fourteen deposition tests (regression sample) were used to fit the regression models used to identify the analytic models required to control the activation strategy implemented in the UEPACTIVATIONVOL user-defined subroutine;Six deposition tests (regression validation sample) were used to validate the regression models previously identified;Seven deposition tests (FE validation sample, Table 2), with increasing laser power (*P*), powder feed rate (*Q*), and scanning speed (*v*) values, were extracted in order to validate the FE model.

### 4.1. Geometry and Mesh

The geometry of the model is illustrated in Figure 5. It consisted of the substrate (grey) with overall dimensions of 10.8×21.6×4.2
mm^3^ and the activation volume (cyan) of 1.2×11×0.6
mm^3^ above the substrate. The substrate represents the bulk material above which the metal powder is fed in order to create the deposited track. During the simulation, only the activation volume was subjected to the activation strategy. The size of the track in fact depends on the process parameters, and only a portion of this additional volume was activated.

The geometry was meshed using an ad-hoc meshing strategy, defined in order to generate a finer mesh in the volume involved in the laser–powder interaction and a coarser mesh outside this volume. Different partitions were generated in the volume, defining three mesh zones, corresponding to the activation volume, the remelted volume inside the substrate, and the substrate (Figure 5). In this way, it was possible to balance the computational time with the resolution of the results, thus making the simulation times compatible with industrial practice, without losing any useful information on the melt pool formation. Moreover, it is useful to point out that mesh size also affects the results of the element activation; in fact, an element that is partially inside the activation volume is only activated if more than one-half of its nodes are included in the activation volume. Meshing operation was performed at the pre-processing phase. There was no remeshing during the calculation process. The structured mesh was generated using Altair HyperMesh™ 2017. The element type used in this work was a 20-node quadratic heat transfer hexahedron (DC3D20). The size of the finest mesh was 0.1
mm, and the mesh size was gradually incremented to 2.7
mm. The model consisted of about 124,000 nodes and about 28,250 elements. The mesh was then imported into Abaqus/CAE.

### 4.2. Material Properties

The material considered in this model was 316L stainless steel. The constant material properties are detailed in Table 3. As a result of a lack of specific results in the literature about solidification of 316L at high cooling rates, the non-equilibrium temperatures were selected according to Mills [57]. The values of the convection coefficient for the deposited track and for the substrate were taken from the experimental results of Gouge et al. [44]. A constant value of 0.6 was adopted for the emissivity to model the thermal radiation as a heat loss mechanism [58]. Figure 6 shows the temperature-dependent properties of the considered material [57], namely density, specific heat, and thermal conductivity. The correction coefficient β, which considers the power reduction due to the interaction with the powder stream, was assumed equal to 0.4, according to Unocic and DuPont [20]. A temperature of 293 K was assumed as the initial temperature $T_{0}$ of the substrate.

### 4.3. Activation Strategy

A regression analysis was performed on the regression sample and used to link the process parameters to the width and the height of the deposited track. The resulting equations are:(12)H=0.0223+0.0015·E+0.0264·G
(13)W=0.3730+0.0049·E−0.0080·G
Equations (Equation 12) and (Equation 13) give a $R_{adj}^{2}$ of 87% and 96%, respectively. Equations (Equation 12) and (Equation 13) were introduced into the UEPACTIVATIONVOL user-defined subroutine in order to define the activation volume. According to the experimental results of Ibarra-Medina and Pinkerton [18], for a laser power of 360 W, the temperature increase was about 36 K. Assuming the powder temperature equal to the initial temperature of the substrate ($T_{0}=293\;K$), the activation temperature $T_{act}$ resulted in 329 K, which was approximated to a value of 350 K for each new activated element.

## 5. Results and Discussion

The regression validation sample was used to assess the forecasting capability of the regression model. Table 4 shows the comparison between the experimental measurements and the height, *H*, and the width, *W*, predicted by the regression model for each set of process parameters. The results indicated that the regression model was able to predict the two dimensions with an average error of about 10%. The maximum error of about 20% on the height was observed for the combination of parameters P=280 W, Q=75 mg/s, and v=15 mm/s.

The regression model was thus introduced in the mesh activation user-defined subroutine, and the results obtained from the FE model are presented in the following sections, in terms of temperature distribution, melt pool geometry, and track dimension. The simulations were carried out using a workstation equipped with an Intel^®^ Core™ i7-6700K processor with 4 GHz and 32 GB of RAM memory using two CPUs (Intel Corporation Italia, Assago, MI, Italy). The mean duration of the simulations was 90 min.

### 5.1. Temperature Distribution

The main output of the model was the temperature distribution. Figure 7 shows the temperature distributions on the top and in two sections obtained from Set 6. Sections were captured at 3/4 of the total length of the deposition path, which was parallel to the Y-axis, where steady-state conditions were reached. The maximum temperature was 2921 K. The sections are plotted considering two different planes: the YZ plane, which was parallel to the deposition direction and located in the middle of the deposition track (Section A–A), and the XZ plane, which was normal to the deposition direction and located at the maximum penetration depth (Section B–B).

The temperature distribution was characterized by the typical double-ellipse shape. The asymmetry of the thermal distribution in the YZ plane was due to the laser beam motion and heat conduction through the bulk material, including the substrate and the deposited track. From the thermal distribution in the XZ plane, it was possible to observe qualitatively that the isotherm curves resembled an arc with the center on the laser beam axis. The axial symmetry, with respect to the YZ plane of the results, was due to the symmetry of the model, since a single track was considered.

The colored Abaqus maps in Figure 8 show the temperature distributions (using the same legend as Figure 7) in the XZ plane for all the sets in Table 2. Using different sets of process parameters, different levels of input energy were obtained, and as a consequence, different temperature distributions were observed. The maximum value of the temperature varied from between about 2077 K for Set 1 to 2450 K for Set 6.

### 5.2. Melt Pool Shape and Dimensions

Table 5 summarizes the experimental values taken from El Cheikh et al. [48] and the numerical results for each single track, in terms of height (*H*), width (*W*), and penetration depth (*D*). A graphic comparison of the numerical and experimental data is presented in Figure 8. The height, the width, and the penetration depth of the melt pool for each set of parameters were measured elaborating the temperature distribution maps using Abaqus/Viewer. In detail, the penetration depth was evaluated as the distance between the top surface of the substrate and the lowest point of the melt pool boundary, according to Figure 9. The melt pool boundary (dotted line superimposed onto the colored Abaqus maps in Figure 8) was evaluated considering the solidus temperature $T_{sol}$. The numerical results were consistent with the experimental results. The model, which used 0.1
mm quadratic mesh elements in the activation volume, was able to forecast the dimensions of the melt pool with an average absolute error of 13% for the height and of 14% for the width. The maximum errors were −0.09
mm for the height (Set 7) and 0.17
mm for the width (Set 3), whereas the maximum percentage errors were −26% for the height (Set 4) and 21% for the width (Set 2). It may be observed that actual DED tracks were generally irregular in profile, and a variation by 5% in height and by 30% in width was common [59,60]. The model also allowed the penetration depth to be predicted with a positive error of less than 0.04
mm, comparable with the mesh resolution, with an average prediction error of 18%.

The numerical results in Figure 8a and Set 1 showed that the process parameters were able to melt the deposited material, but a negligible penetration depth was observed. Thus, the deposit failed. The results of the FE model were consistent with those of the experimental campaign, which also exhibited an issue of adhesion. Overall, the results proved that the model was capable of identifying the combination of process parameters that can generate a reliable adhesion between the deposited material and the substrate. In fact, a loss of adhesion could lead to detachments, which could in turn negatively affect the functionality of fabricated components and lead to a waste of time and materials.

Figure 8a shows the effect of laser power on the melt pool dimensions. As the thermal power increased, the thermal energy available at the substrate increased, and consequently, the penetration depth also increased. As expected, it was possible to observe an increase in the track width as the laser power increased, even though the numerical model overestimated the increase. The height of the track was less dependent on the variation of the laser power.

Figure 8b shows the variations of the melt pool dimensions as a function of the scanning speed. As the scanning speed increased, the height of the deposited material decreased, and consequently, the thermal energy of the material deposited above the substrate decreased. Moreover, the thermal energy supplied by the laser also decreased, because the thermal power was applied over a smaller time interval. In LP-DED, a certain part of the energy is initially used to melt the powder and the substrate, and the remaining part has the effect of increasing the penetration depth. The melting rate was very rapid, up to 9000 K/s, while the temperature penetration was a function of the thermal properties of the material and the laser-material interaction time. A shorter time interval thus implied a reduction in the penetration depth. However, the dependence of penetration depth from interaction time was not linear, since reducing the time, the process conditions were nearer the threshold, and the initial melting prevailed on the temperature propagation. In the experiments, if the scanning speed exceeded a value of 10 mm/s, the penetration depth remained constant, and this behavior was well predicted by the developed hybrid model.

Figure 8c shows the correlations between the track geometry and the powder feed rate. When the powder feed rate was increased, the height of the deposited track increased according to Equation (Equation 12), and consequently, the mass of hot deposited material present on the substrate also increased. The increase in volume of the hot deposited material caused an increase in the penetration depth due to thermal inertia. If the powder feed rate increased above a certain value, that is more than 50 mg/s for this set of experiments, the volume of new deposited (activated) material increased, and more energy was needed to melt it. As a result, since the laser power and the scanning speed were constant, less energy was made available to the substrate, which led to a reduction in the penetration depth.

## 6. Conclusions

A thermal LP-DED model, developed using the finite element method, was proposed in this work. The model took into account the main physical characteristics of the process, such as the heat source distribution, the convective heat transfer due to the carrier gas, the thermal radiation, and the convective flows into the melt pool. All the assumptions of the implemented model were based on experimental knowledge retrieved from the literature. A specific mesh strategy, with a finer mesh in the melt pool area, allowed the melt pool generation phenomenon to be investigated in detail. A new activation strategy was used to simulate the addition of material during the process. The volume of the activated elements depended on the process parameters that were introduced into the simulation model. The main results and conclusions that can be drawn from the present work are:a novel material addition strategy was adopted, in which the volume of the deposited material was a function of process parameters and was modeled by combining regression-based analytic models with an element activation approach in the Abaqus software tool;a comparison with experimental data from the literature showed that the the model was able to predict the height and the width of the deposited track with an average percentage error of 14% and a maximum deviation of 0.17
mm, as well as the penetration depth with an average percentage error of 18% and a maximum deviation of 0.04
mm; the forecasting accuracy was consistent with the adopted mesh resolution;by comparing the proposed model with other models available in the literature, the prediction error was slightly higher, being approximately 10%, the typical declared error for the track height; as concerns the track width, the prediction error was within the variability of actual tracks;the model was thus able to capture the main aspects of the process physics and to make a reasonable short-time forecast of the results, in terms of track geometry and temperatures;the model allowed recognizing whether a set of process parameters would ensure adhesion between the deposited material and the substrate; this allowed the limits of the process parameter window to be identified, and thus saving time and the cost of a complete experimental campaign.

## Figures and Tables

**Figure 1 materials-12-02819-f001:**
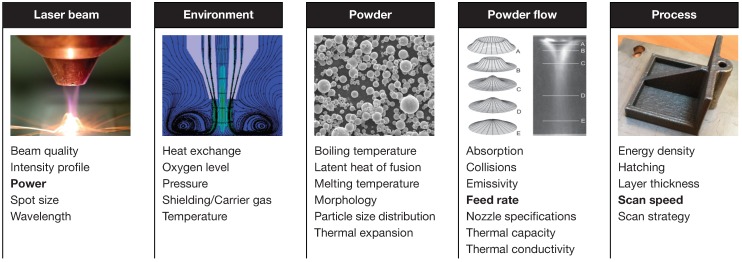
Main parameters that affect the Laser Powder Directed Energy Deposition (LP-DED) process.

**Figure 2 materials-12-02819-f002:**
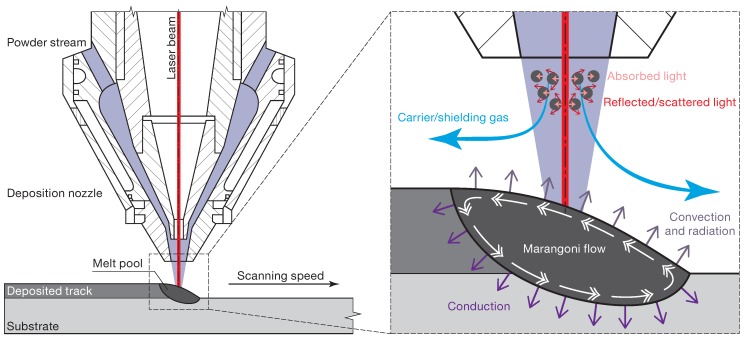
Schematic representation of the main phenomena involved in the LP-DED process.

**Figure 3 materials-12-02819-f003:**
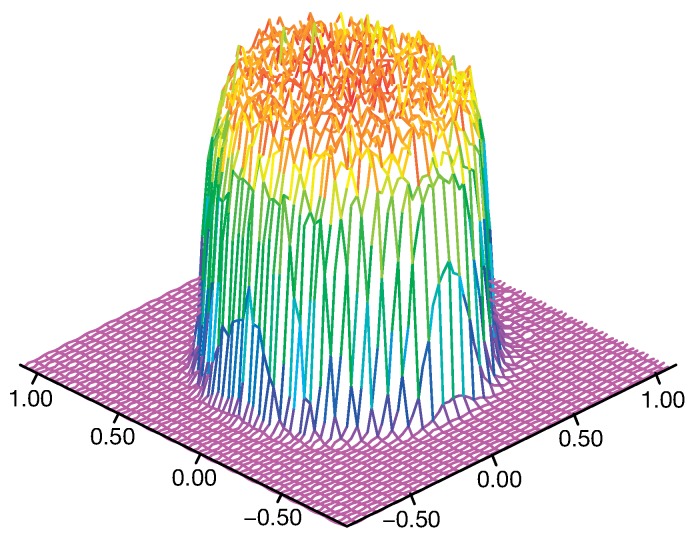
Isometric representation of a laser beam intensity distribution.

**Figure 4 materials-12-02819-f004:**
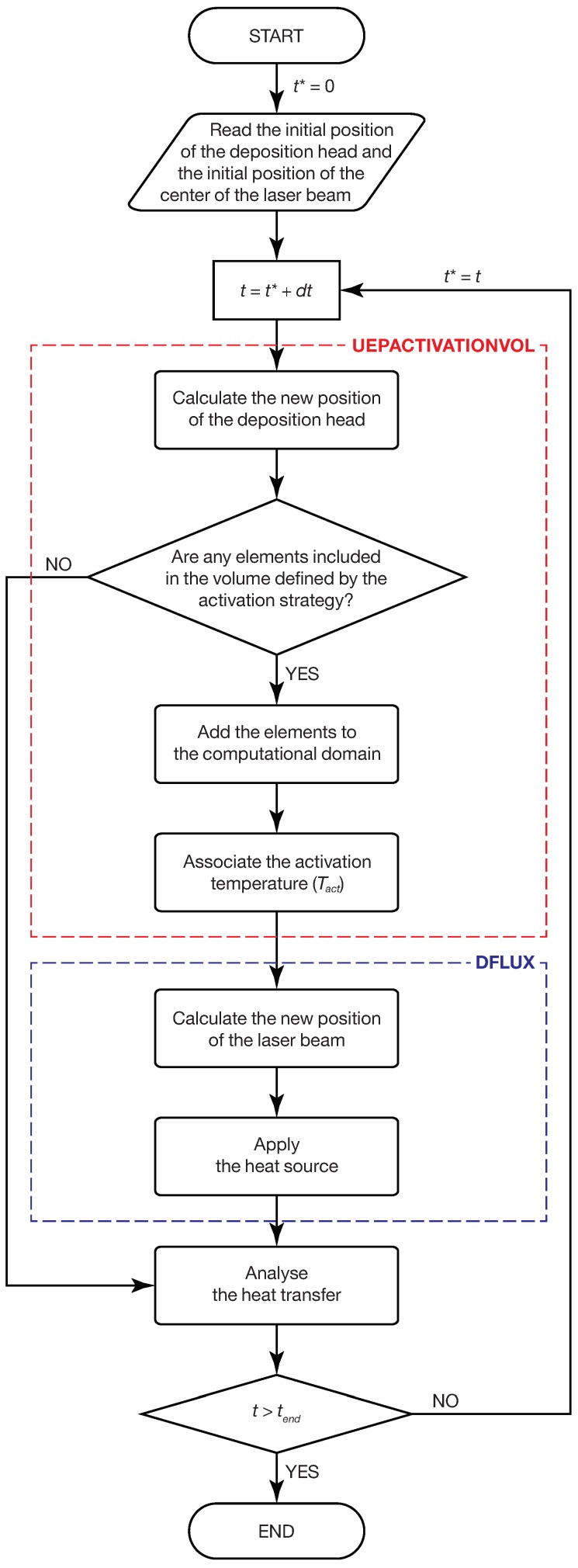
Flowchart of the numerical model.

**Figure 5 materials-12-02819-f005:**
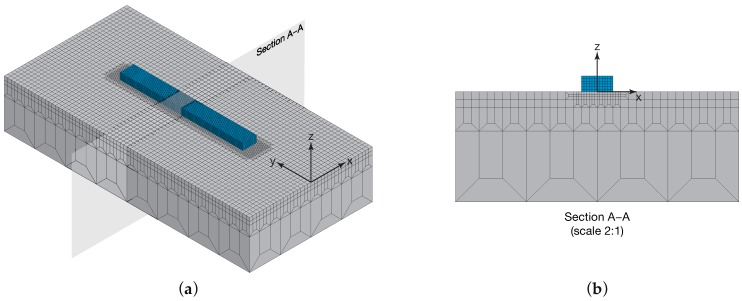
The geometry of the model consists of the substrate (grey) with overall dimensions of 10.8×21.6×4.2
mm^3^ and the activation volume (cyan) of 1.2×11×0.6
mm^3^ above the substrate. (**a**) Isometric view of the finite element model and (**b**) its cross-section on the XZ plane.

**Figure 6 materials-12-02819-f006:**
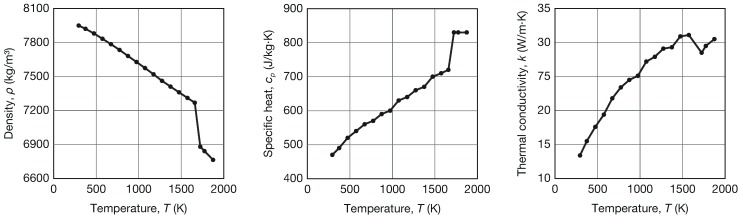
Temperature dependent thermal properties of 316L stainless steel [57].

**Figure 7 materials-12-02819-f007:**
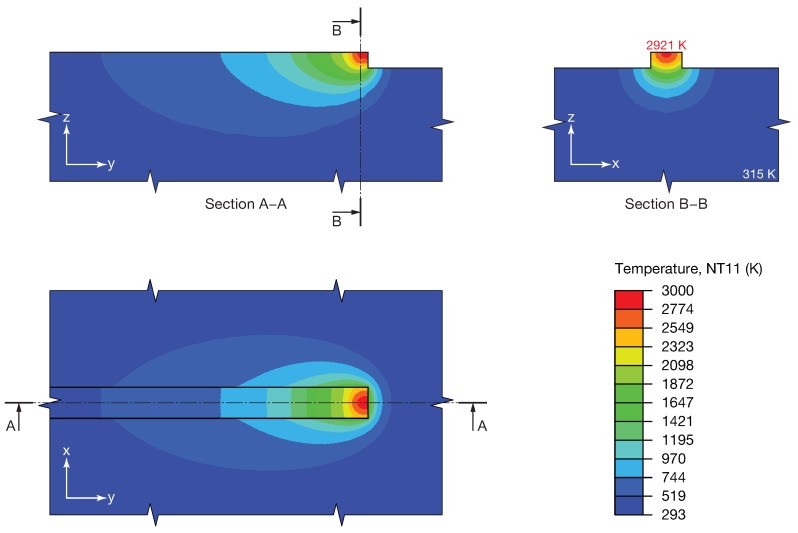
Temperature distributions on the top and in two cross-sections for Set 6.

**Figure 8 materials-12-02819-f008:**
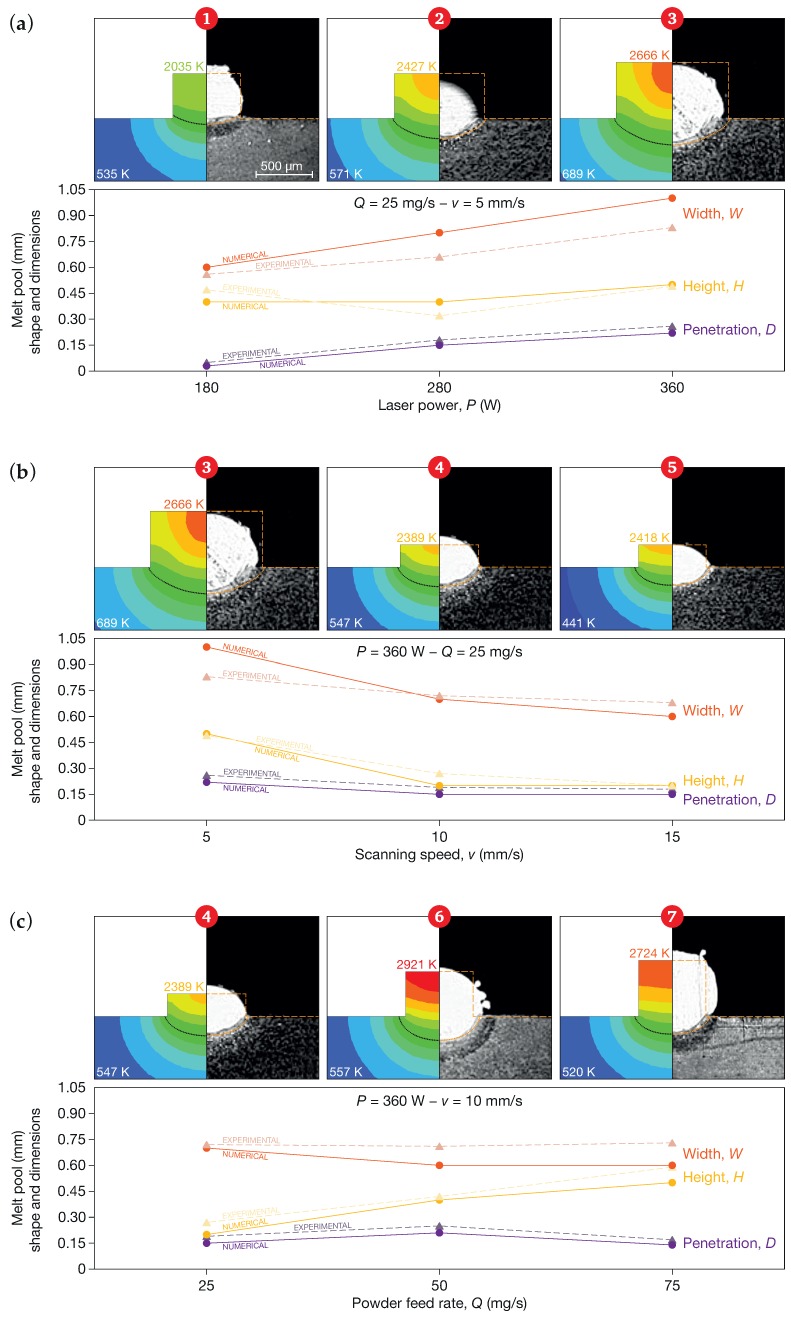
Temperature distributions and melt pool dimensions: comparison of the numerical results with experimental data as a function of (**a**) laser power, (**b**) scanning speed, and (**c**) powder feed rate. The temperature legend is the same as that of Figure 7, and for each set, the maximum and minimum temperatures are shown. The dotted line represents the melt pool boundary.

**Figure 9 materials-12-02819-f009:**
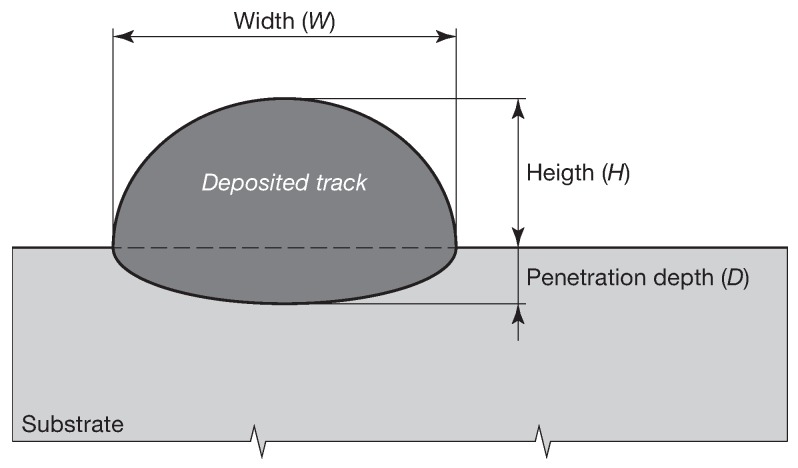
Schematic of measures for each single track.

**Table 1 materials-12-02819-t001:** Summary of the LP-DED models available in the literature.

Ref.	Materials	Model Output	Assumptions
[25]	316L	Temperature distribution	No heat source No convection or radiation Substrate at a constant temperature Elements activated at the melting temperature
[26]	Mild steel	Melt pool dimensions	Gaussian distribution of the heat source Latent heat included by a modification of the specific heat Combined heat transfer coefficient No Marangoni flows The addition of material is neglected
[29]	400 Monel alloy	Temperature distribution Melt pool dimensions Residual stress distribution	Gaussian distribution of the heat source Latent heat included by a modification of the specific heat Combined heat transfer coefficient No Marangoni flows
[30]	Iron	Effect of laser pulsed shape parameters on the geometryof the deposited material	Gaussian distribution of the heat sourceLatent heat included by a modification of the specific heat Combined heat transfer coefficient
[31]	Iron	Effect of powder feed rate and scanning speed on the geometryof the deposited material	Gaussian distribution of the heat source Latent heat included by a modification of the specific heat Combined heat transfer coefficient
[32]	420 steel	Microstructure Hardness	Gaussian distribution of the heat source Convection coefficient = 30 W/($m^{2}$·K) An internal heat source simulates the latent heat Elements activated at their melting temperature
[33]	SS410	Solid phase evolution	Gaussian distribution of the heat source
[34]	Ti-6Al-4V	Temperature distribution in the substrate	Heat source applied to the substrate Convection coefficient = 20 W/($m^{2}$·K) Addition of material simulated by a modification of the thermal conductivity
[35]	304 L	Track geometry Temperature distribution in the substrate	Gaussian distribution of the heat source Latent heat included by a modification of the specific heat Heat source applied to the substrate Convection coefficient = 40 W/($m^{2}$·K) Elements activated at ambient temperature
[36]	SS316	Temperature distribution	Gaussian distribution of the heat source Convection coefficient = 10 W/($m^{2}$·K) Initial temperature of the entire model = 300 K
[14]	Ti-6Al-4 V	Temperature distribution	Track geometry No radiation or convection Gaussian distribution of the heat source No Marangoni flows
[37]	316L	Peak temperature/cooling rate Melt pool geometry	Gaussian distribution of the heat source No radiation
[38]	316L	Temperature distribution Re-melted depth	Natural convection Gaussian distribution of the heat source
[39]	Ti-6Al-4V	Temperature distribution in the substrate	Combined heat transfer coefficient Heat source applied as a constant energy density Model calibration is required
[40]	Ti-6Al-4V Inconel 718	Residual stress distribution	Rectangular laser beam shape No radiation Convection coefficient = 20 W/($m^{2}$·K) No Marangoni flows
[22]	Co-base alloy	Temperature distribution	Gaussian distribution of the heat source No energy attenuation Convection coefficient = 100 W/($m^{2}$·K)

**Table 2 materials-12-02819-t002:** Process parameter sets for each single track used for the model validation.

Set	*P*	*Q*	*v*
	(W)	(mg/s)	(mm/s)
1	180	25	5
2	280	25	5
3	360	25	5
4	360	25	10
5	360	25	15
6	360	50	10
7	360	75	10

**Table 3 materials-12-02819-t003:** Constant thermal properties of 316L stainless steel [57,58].

Property	Value
Solidus temperature, $T_{sol}$	1674 K
Liquidus temperature, $T_{liq}$	1697 K
Latent heat of fusion, $L_{f}$	$2.6\times 10^{4}$J/kg
Surface emissivity coefficient, $\varepsilon_{r}$	0.6

**Table 4 materials-12-02819-t004:** Track dimensions predicted by the regression model and comparison with experimental data.

*P*	*Q*	*v*	$H_{\exp}$	$H_{\mathrm{reg}}$	ΔH		$W_{\exp}$	$W_{\mathrm{reg}}$	ΔW	
**(W)**	**(mg/s)**	**(mm/s)**	**(mm)**	**(mm)**	**(mm)**	**(%)**	**(mm)**	**(mm)**	**(mm)**	**(%)**
180	75	10	0.47	0.45	-0.02	−5%	0.39	0.43	+0.04	9%
280	50	10	0.37	0.35	-0.02	−5%	0.62	0.56	-0.06	−10%
280	75	5	0.80	0.93	+0.13	16%	0.65	0.67	+0.02	3%
280	75	15	0.27	0.32	+0.05	20%	0.56	0.47	-0.09	−16%
360	50	5	0.82	0.72	-0.10	−12%	0.86	0.89	+0.03	4%
360	50	15	0.26	0.26	+0.00	0%	0.62	0.55	-0.07	−12%

**Table 5 materials-12-02819-t005:** Experimental measurements and numerical results for each single track.

Set	$H_{\exp}$	$H_{\mathrm{num}}$	ΔH		$W_{\exp}$	$W_{\mathrm{num}}$	ΔW		$D_{\exp}$	$D_{\mathrm{num}}$	ΔD	
	(mm)	(mm)	(mm)	(%)	(mm)	(mm)	(mm)	(%)	(mm)	(mm)	(mm)	(%)
1	0.47	0.40	-0.07	−15%	0.56	0.60	0.04	7%	0.03	0.05	0.02 †	67% †
2	0.32	0.40	0.08	25%	0.66	0.80	0.14	21%	0.15	0.18	0.03	20%
3	0.49	0.50	0.01	2%	0.83	1.00	0.17	20%	0.22	0.26	0.04	18%
4	0.27	0.20	-0.07	−26%	0.72	0.70	-0.02	−3%	0.15	0.19	0.04	27%
5	0.20	0.20	0.00	0%	0.68	0.60	-0.08	−12%	0.15	0.18	0.03	20%
6	0.42	0.40	-0.02	−5%	0.71	0.60	-0.11	−15%	0.21	0.25	0.04	19%
7	0.59	0.50	-0.09	−15%	0.73	0.60	-0.13	−18%	0.14	0.17	0.03	21%

† No adhesion of the deposit to the substrate; consequently, the value of the error is not significant.

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
