# Peer review of "A Hybrid Modeling of the Physics-Driven Evolution of Material Addition and Track Generation in Laser Powder Directed Energy Deposition"

_materials, 2019, doi:10.3390/ma12172819_

Round 1

Reviewer 1 Report

This work presents a thermal modeling work for directed energy deposition additive manufacturing implemented in  Abaqus/Standard 2018. There is a nice literature review, and a clear presentation of the fundamentals of the implementation including governing equations. The model is then run for a simple single pass scan with various parameters and is compared to experimental results showing reasonable comparisons to deposition bead dimensions. The work has merit and is well done, however several issues should be addressed prior to publication. The authors should be commended for using experimental data to first calibrate the model, then additional experimental data points to compare to model predictions.

Major issues:

In the element activation subroutine it appears based on the text that activated elements are deposited at an initial melt temperature. Then in the next step laser heating is applied to the activated elements. Would this not input too much energy since the laser heating is also responsible for the energy to bring powder particles to melt? The results do seem reasonable so perhaps clarification would help.

The activation strategy presented is inherently mesh dependent, thus allowable deposition dimensions are limited to discrete values based on mesh resolution. This makes validation of the model based on deposition dimensions somewhat suspect. The authors mention some sort of mesh refinement scheme and uncertainties in the reported dimensions, yet exactly what this refinement looks like is not clear. Perhaps a picture of the refinement would be helpful.

Minor Issues:

Figure 9 should be referenced in the text.

Line 94: melting is described as happening instantaneously, yet surely the heat transfer takes place over some sort of discrete time. Perhaps the wording should be, 'rapidly' or, 'very rapidly?'

Line 195: should be, 'quiet' instead of 'quite'

Line 196: Please be more specific as to exactly what peoperties of the elements 'main properties' refers to. Material properties, Integration schemes, etc. ?

Author Response

Dear Reviewer,

we are very grateful to the Reviewer for his/her general appreciation of our work, as well as for his/her valuable suggestions to improve the quality and the impact of our contribution.

We considered every single comment raised by the Reviewer and a point-by-point reply is reported in the attachment. Moreover, all the changes are tracked in the revised manuscript.

Best regards,
Alessandro Salmi

Reviewer 2 Report

The manuscript materials-579264 entitled “A hybrid modelling of the physics-driven evolution of material addition and track generation in laser powder directed energy deposition (LP-DED)” presents an interesting numerical study applied to additive manufacturing. Overall the paper is well-written and organized, with an extensive and thorough literature review but there are some issues:

·         Please define all the acronyms first time they are employed, e.g. AM in the abstract

·         please be consistent with units e.g. in line 282 and 299 “g/s” and “mg/s”

·         Please be careful with some terminology and use always suitable terms e.g. “fabricated” in line 360.

·         It is fine to use such a meshing strategy as ad-hoc, but did the authors tried other strategies?

·         “Involved” in line 369

·         Why is the mesh generated in another software? I am familiar with both commercial codes and I know from personal experience that HM has powerful mesh generator algorithms but this one looks like an easy one to model in Abaqus. Is there any specific reason behind that?

·         Regarding material properties, what about other properties related for instance with material expansion?

·         Line 396, please keep the results in its section.

·         Line 399, please give more details regarding the activation temperature. 350 K is a low value in my opinion. This choice must be explained and supported.

·         Figure 7 shows one of the extremities at room temperature for ¾ of the path. Is the cooling so rapid, considering the high values for scanning speed?

·         Please provide the subroutines used.

·         Please provide the maximum temperature for each set.

·         In section 5.2 and table 4, please provide the error in percentage for the prediction of the dimensions of the melt pool. From figure 8, it is possible to verify a significant error in the width prediction.

·         Why was the track defined as a rectangular section instead of an ellipse defined also by 2 parameters?

·         “If the scanning speed exceeds a value of 600mm/min, the penetration depth remains constant, and this behaviour is well predicted by the developed model.” Please provide a further explanation for the independence of the penetration depth for scanning speeds over 600 mm/min.

·         Please highlight in the conclusions the novelty of this model in comparison with others reviewed in the literature.

·         Line 481: “a comparison with experimental data from the literature has shown a significant forecasting accuracy of the model, with a percentage error of 20% and a maximum deviation of 0.04mm for the prediction of the penetration depth.” 20% is a significant error, therefore this type of statements must be revised. Additionally, please compare these error values with other models available in the literature.

Author Response

Dear Reviewer,

we are grateful to the Reviewer for his/her general appreciation of our work and for the precious hints and suggestions, which gave us the opportunity to improve our contribution.

We addressed every suggested revision to thoroughly and we provide a point-by-point reply in the attachment. Moreover, all the changes are tracked in the revised manuscript.

Best regards,
Alessandro Salmi

Round 2

Reviewer 2 Report

Dear Authors,

Thank you for thoroughly addressing all my suggestions. I have nothing to add.

Best regards